# Sulodexide Prevents Hyperglycemia-Induced Endothelial Dysfunction and Oxidative Stress in Porcine Retinal Arterioles

**DOI:** 10.3390/antiox12020388

**Published:** 2023-02-06

**Authors:** Alice Dauth, Andrzej Bręborowicz, Yue Ruan, Qi Tang, Jenia K. Zadeh, Elsa W. Böhm, Norbert Pfeiffer, Pratik H. Khedkar, Andreas Patzak, Ksenija Vujacic-Mirski, Andreas Daiber, Adrian Gericke

**Affiliations:** 1Department of Neurosurgery, University Medical Center, Johannes Gutenberg University, Langenbeckstr. 1, 55131 Mainz, Germany; 2Department of Ophthalmology, University Medical Center, Johannes Gutenberg University, Langenbeckstr. 1, 55131 Mainz, Germany; 3Department of Pathophysiology, Poznań University of Medical Sciences, 60-512 Poznań, Poland; 4AbbVie Germany GmbH & Co. KG, 65189 Wiesbaden, Germany; 5Institute of Translational Physiology, Charité-Universitätsmedizin Berlin, Corporate Member of Freie Universität Berlin and Humboldt-Universität zu Berlin, Charitéplatz 1, 10117 Berlin, Germany; 6Department of Cardiology, Cardiology 1, University Medical Center, Johannes Gutenberg University, Langenbeckstr. 1, 55131 Mainz, Germany; 7Partner Site Rhine-Main, German Center for Cardiovascular Research (DZHK), 55131 Mainz, Germany

**Keywords:** diabetic retinopathy, endothelial dysfunction, oxidative stress, sulodexide

## Abstract

Diabetes mellitus may cause severe damage to retinal blood vessels. The central aim of this study was to test the hypothesis that sulodexide, a mixture of glycosaminoglycans, has a protective effect against hyperglycemia-induced endothelial dysfunction in the retina. Functional studies were performed in isolated porcine retinal arterioles. Vessels were cannulated and incubated with highly concentrated glucose solution (HG, 25 mM D-glucose) +/− sulodexide (50/5/0.5 μg/mL) or normally concentrated glucose solution (NG, 5.5 mM D-glucose) +/− sulodexide for two hours. Endothelium-dependent and endothelium-independent vasodilatation were measured by videomicroscopy. Reactive oxygen species (ROS) were quantified by dihydroethidium (DHE) fluorescence. Using high-pressure liquid chromatography (HPLC), the intrinsic antioxidant properties of sulodexide were investigated. Quantitative PCR was used to determine mRNA expression of regulatory, inflammatory, and redox genes in retinal arterioles, some of which were subsequently quantified at the protein level by immunofluorescence microscopy. Incubation of retinal arterioles with HG caused significant impairment of endothelium-dependent vasodilation, whereas endothelium-independent responses were not affected. In the HG group, ROS formation was markedly increased in the vascular wall. Strikingly, sulodexide had a protective effect against hyperglycemia-induced ROS formation in the vascular wall and had a concentration-dependent protective effect against endothelial dysfunction. Although sulodexide itself had only negligible antioxidant properties, it prevented hyperglycemia-induced overexpression of the pro-oxidant redox enzymes, NOX4 and NOX5. The data of the present study provide evidence that sulodexide has a protective effect against hyperglycemia-induced oxidative stress and endothelial dysfunction in porcine retinal arterioles, possibly by modulation of redox enzyme expression.

## 1. Introduction

Diabetic retinopathy is a leading cause of severe visual impairment and blindness worldwide [1,2,3]. A crucial factor in the pathogenesis of diabetic retinopathy is hyperglycemia-induced damage to the retinal vascular endothelium [2,3,4]. Hyperglycemia causes endothelial dysfunction through oxidative stress, inflammatory processes, glycocalyx damage, cell senescence, and osmotic effects [2,5,6]. Hence, endothelial responses to physiological stimuli like shear stress or vasoactive substances are mitigated [2]. Oxidative stress, which is characterized as an imbalance between extensive reactive oxygen species (ROS) production and antioxidant availability, is a fundamental process in the pathogenesis of diabetic retinopathy [7,8]. Systemic oxidative stress and low-grade inflammation are also hallmarks of diabetes in general [9]. In this context, the enzyme family of NADPH-oxidases [10] is known to importantly contribute to ROS generation in retinal endothelial cells and is essentially involved in the development of vasculopathy in diabetic retinopathy [5,10,11]. According to current research, the superoxide anion (O_2_^•−^) is largely responsible for hyperglycemia-associated endothelial dysfunction by reducing the bioavailability of the vasorelaxing and endothelium-protective factor nitric oxide (NO) via uncoupling of endothelial nitric oxide synthase (eNOS), thereby inhibiting protective mechanisms of endothelial cells and, thus, enhancing inflammatory and cell-damaging processes [12,13]. Consequently, oxidative stress and endothelial dysfunction significantly promote the development of diabetic-retinopathy-associated complications, such as retinal ischemia, edema, retinal or vitreous hemorrhage, and retinal detachment. Common therapeutic strategies include reduction of glucose levels, laser coagulation, or intravitreal application of substances blocking the effects of vascular endothelial growth factor (VEGF). However, apart from therapeutic glucose level reduction, there are so far no established pharmacological therapies preventing the onset or progression of diabetic retinopathy [1,14].

Based on studies in various organs from patients and animals, sulodexide, a mixture of glycosaminoglycans composed of low molecular weight heparin and dermatan sulfate, represents a promising therapeutic target in various vascular damaging diseases [15,16]. Sulodexide has a longer half-life than low-molecular-weight heparins and, in addition, a pronounced oral absorption potential. Sulodexide is currently approved in some parts of Eastern Europe, Africa, and Asia for the treatment of chronic venous insufficiency, for prevention of recurrent venous thromboembolism, and for treatment of diabetic nephropathy [17,18]. In human and animal studies, the compound already revealed endothelium-protective effects in diabetic nephropathy and in ischemia–reperfusion damage of the heart and kidneys [15,16]. Furthermore, sulodexide reduced oxidative stress and had a protective effect against hyperglycemia-induced damage in arterial and venous endothelial cell cultures [19,20,21]. Notably, sulodexide was reported to strengthen the glycocalyx of retinal arterioles in diabetic patients, to reduce vascular permeability, and to suppress retinal neovascularization in vivo, suggesting that sulodexide may be a promising agent for the treatment of diabetic retinopathy [22,23]. Moreover, sulodexide was shown to reduce glucose-induced senescence in human retinal endothelial cells in a recent study [6]. A major aim of the present study was to test the hypothesis that sulodexide has a protective effect against hyperglycemia-induced endothelial dysfunction in retinal arterioles. For that reason, we used an in vitro model, in which porcine retinal arterioles were exposed to high glucose (HG) concentrations and to different concentrations of sulodexide. Another objective of this study was to determine the potential molecular mechanisms activated by HG and sulodexide exposure.

## 2. Materials and Methods

### 2.1. Animals

No experiments on living animals were carried out in this study. Eyes were obtained from castrated male (50%) or female (50%) domestic pigs (*Sus scrofa domesticus*) from a regional farm after they had been killed for consumption purposes. The animals were six to seven months old and had an average weight of 125 kg. Immediately after electrically induced cardiac arrest and exsanguination, the eye globes were excised and placed in ice-cold artificial cerebrospinal fluid (aCSF) of the following ionic composition (in mM; salts obtained from Carl Roth GmbH, Karlsruhe, Germany): 125 NaCl, 2.5 KCl, 1 MgCl_2_, 1.6 NaH_2_PO_4_, 25 NaHCO_3_, 2.5 CaCl_2_ × 2 H_2_O, 5.5 D-glucose.

### 2.2. Functional Studies in Retinal Arterioles

After no longer than 30 min of transport in cold aCSF, the eye globes were opened with a scalpel along the limbus, and the vitreous was removed with tweezers. First-order arterioles were then isolated from the retina under a dissection microscope (Olympus SZ61, Olympus, Shinjuku, Tokyo, Japan) using micro-scissors, and subsequently, retinal tissue was gently cleaned off the arterioles with fine-point tweezers. Small vessel segments (approximately 2 mm in length) were transferred to a perfusion chamber (Jim’s Instrument Manufacture Inc., Iowa City, IA, USA), cannulated by two opposite micropipettes, and fixed with two sutures (10-0 nylon monofilament). Next, the vessels were set to an intraluminal pressure of 40 mmHg via the micropipettes under no-flow conditions. The perfusion chamber was continuously circulated with oxygenated (95%) and carbonated (5%) aCSF-buffer solution (37 °C, pH 7.4). Measurements of vascular reactivity were recorded by videomicroscopy. Viability of vessels was tested with 100 mM KCl. Retinal arterioles that responded to 100 mM KCl with a reduction of luminal vessel diameter by more than 30% were included in the study The vessel lumen was flushed and then incubated for two hours with HG 25 mM (Dulbecco’s Modified Eagle’s Medium, High Glucose, 25 mM, Thermo Fisher Scientific, Waltham, MA, USA) under constant aCSF-circulation in our setup [24]. For further experiments, sulodexide (Vessel Due F 300 LSU/mL, Alfa Wassermann S.p.A, Bologna, Italy) was added for incubation in the following concentrations: 50 µg/mL or 5 µg/mL or 0.5 µg/mL (diluted in HG or NG). The control groups were incubated with NG 5 mM +/− sulodexide (Dulbecco’s Modified Eagle’s Medium, Low Glucose, 5 mM, Thermo Fisher Scientific, Waltham, MA, USA). After two hours of incubation, vascular reactivity to the thromboxane mimetic 9,11-dideoxy-9 α,11α-methanoepoxy prostaglandin F2α (U44619, 10^−11^–10^−6^ M, Cayman Chemical Company, MI, USA) was measured to test whether incubation with HG and/or sulodexide induced changes in smooth muscle contractility. Then, endothelium-dependent vasodilation to bradykinin (10^−12^–10^−7^ M, Sigma-Aldrich, Steinheim, Germany) and endothelium-independent vasodilatation to sodium nitroprusside (SNP; 10^−9^–10^−4^ M, Sigma-Aldrich) was measured. The preconstriction level was set to 50–70% of the initial diameter by titration of U46619 if the spontaneous myogenic tone was too weak [25]. Stock solutions were dissolved in PBS or dimethyl sulfoxide (DMSO, Thermo Fisher Scientific, Waltham, MA, USA).

### 2.3. Detection of ROS in Retinal Arterioles

Porcine retinal pieces containing a first-order arteriole were freshly isolated. Next, the proximal end of the arteriole was cannulated with a micropipette, and the whole vascular tree was perfused with DMEM containing HG or NG with either sulodexide or vehicle solution (saline), respectively. Next, the retinal piece was transferred to a petri dish and incubated in the respective solution at 37 °C for two hours. After incubation, retinal arterioles were isolated, positioned in cryomolds containing Tissue-Tek O.C.T. compound (both Sakura Finetek Europe, Alphen aan den Rijn, The Netherlands), and snap-frozen in liquid nitrogen. Cryosections (10 µm thickness) (Leica Reichert Jung, 2030; Leica, Rijswijk, The Netherlands) were placed on glass slides (SuperFrost Plus™; Thermo Fisher Scientific, Waltham, MA, USA) and incubated with dihydroethidium (DHE, 1 µM, Thermo Fisher Scientific, Waltham, MA, USA) for 30 min at 37 °C. Fluorescent intensity was recorded using a fluorescence microscope (Eclipse TS 100; Nikon, Yurakucho, Tokyo, Japan) equipped with a DS-Fi1-U2 digital microscope camera (Nikon) and an ELWD 20 ×/0.45 S Plan Fluor Ph1 ADM objective (Nikon). Subsequently, imaging was performed using NIS Elements imaging software (Nikon, version 1.10 64 bit; setting: TRITC filter, exposure 330 ms, enhancement 2×), then fluorescence intensity was measured by using ImageJ image-processing software (freeware, developer Wayne Rasband, National Institutes of Health, Bethesda, USA), as described previously [26].

### 2.4. HPLC-Based Assays to Assess the Antioxidant Properties of Sulodexide

The direct antioxidant properties of sulodexide (1–1000 µM) were tested by three different HPLC-based assays. The HPLC system was purchased from Jasco, Groß-Umstadt, Germany, with a typical composition: control unit, degasser unit, two pumps for high pressure gradient, high pressure mixer, UV/Vis and fluorescence detectors, and an autosampler (AS-2057) with a 4 °C cooling device. For separation of all products and reactants, a reversed-phase column was used (C18-Nucleosil 100-3, 125 mm × 4 mm, Macherey & Nagel, Düren, Germany), together with solvents A (50 mM citric acid in water, pH 2.2) and B (acetonitrile with 10 *v/v*% water). Superoxide scavenging was tested by incubation of increasing concentrations of sulodexide with xanthine oxidase (20 mU/mL)/1 mM hypoxanthine to generate superoxide and hydroethidine (DHE, 50 µM) as a superoxide probe in phosphate-buffered saline pH 7.4 at 37 °C for 30 min. In total, 50 µL of the reaction solution was subjected to HPLC analysis. The oxidation products, 2-hydroxyethidium (specific for superoxide) and ethidium (unspecific for ROS and RNS), were separated and quantified by a reversed-phase HPLC protocol coupled with fluorescent detection, as reported [27].

Peroxynitrite and derived free radical scavenging was tested by incubation of increasing concentrations of sulodexide with peroxynitrite (250 µM, synthesized as described [28]) and phenol (5 mM) as a target of nitration in 0.1 M potassium phosphate buffer pH 7.4 at 37 °C for 5 min. In total, 50 µL of the reaction solution was subjected to HPLC analysis. The nitration products, p- and o-nitrophenol, were separated and quantified by a reversed-phase HPLC protocol coupled with UV/Vis detection that was adapted from a previously published protocol [29]. Briefly, separation was achieved by application of an isocratic solvent system consisting of 70% solvent A and 30% solvent B. Between 10 and 12 min, a washing gradient was applied with a peak percentage of 90% solvent B. The flow was 1 mL/min, and the nitrated products were detected by their absorption at 287 nm (but also wavelengths 300 and 350 nm were monitored). Typical retention times of the products, p-nitrophenol and o-nitrophenol, were 4.55 min and 6.79 min, respectively. Typical retention time of the substrate, phenol, was 3.16 min. Total time of the HPLC method was 13 min.

Hydrogen peroxide scavenging was tested by incubation of increasing concentrations of sulodexide with hydrogen peroxide (250 µM) and ebselen (250 µM) as a target of oxidation in phosphate-buffered saline pH 7.4 at 37 °C for 90 min. In total, 50 µL of the reaction solution was subjected to HPLC analysis. The oxidation product and the substrate, ebselen-oxide and ebselen, were separated and quantified by a reversed-phase HPLC protocol coupled with UV/Vis detection that was adapted from a previously published protocol [30]. Briefly, separation was achieved by application of a high pressure gradient. The following percentages of the organic solvent were applied: 0–10 min, 30 to 70% B. Between 10 and 12 min, a washing gradient was applied with a peak percentage of 90% solvent B. The flow was 1 mL/min, and ebselen-oxide was detected by its absorption at 270 nm (but also wavelength 300 nm was monitored). Typical retention time of the product, ebselen-oxide, was 2.80 min. Typical retention time of the substrate, ebselen, was 7.08 min. Total time of the HPLC method was 14 min.

### 2.5. Quantitative PCR Analysis (qPCR)

Retinal pieces were isolated, perfused, and incubated for two hours, as described for DHE staining. After incubation, retinal pieces were washed in PBS, and retinal arterioles were isolated. Next, arterioles were washed in PBS, cut into small pieces, placed into 1.5 mL reaction tubes (Eppendorf Vertrieb GmbH, Hamburg, Germany), and snap-frozen. Later, RNA was extracted from the isolated retinal vessels using the E.Z.N.A. total RNA Kit (Omega Bio-Tek, Norcross, GA, USA). Reverse transcription of total RNA (1 μg) was performed using the High-Capacity cDNA Reverse Transcription Kit (Applied Biosystems, Foster City, CA, USA). Then, qPCR was performed in an iCycler iQ system (Bio-Rad Laboratories, Munich, Germany) using the Absolute qPCR SYBR Green Fluorescein Mix Kit (Thermo Fisher Scientific, Surrey, the UK). The primer sequences are shown in Table 1. The comparative threshold cycle method was used for relative mRNA quantification [31]. Relative mRNA levels of target genes were quantified using a comparative threshold normalized to the β*-actin* gene (*ACTB*). Primer sequences are presented in Table 1.

### 2.6. Immunostainings

Cryosections (10 µm thickness) with retinal arteriole cross-sections were fixed with 4% PFA (paraformaldehyde; Carl Roth GmbH, Karlsruhe, Germany), washed in PBS and incubated with blocking medium (10 mL PBS + 0% Triton X-100 + 1% BSA; Sigma-Aldrich^®^, Steinheim, Germany). Next, primary rabbit polyclonal antibodies targeted against NOX2 (Abcam, Waltham, MA, USA, catalog number: ab80508, dilution 1:200), NOX4 (Abcam, catalog number: ab154244, dilution 1:200), and NOX5 (Abcam, catalog number: ab191010, dilution 1:200) were applied on the tissue sections and incubated for two hours at room temperature. A Rhodamin Red-X-coupled, goat anti-rabbit polyclonal secondary antibody (Dianova GmbH, Hamburg, Germany, catalog number: 111-295-003; dilution 1:200, incubation time: 1 h at RT) was used. Negative control sections were incubated with secondary antibody only. Finally, the slides were mounted using VECTASHIELD^®^ Mounting Medium with 4′,6-diamidin-2-phenylindol (DAPI Vectashield, BIOZOL Diagnostica Vertrieb GmbH, Eching, Germany) and covered with coverslips. Fluorescence intensity was visualized (Filter DAPI, exposure 300 ms; filter TRITC, exposure 300 ms) and quantified as described previously [26].

### 2.7. Statistical Analysis

Concentration–response curves were compared by two-way analysis of variance (ANOVA) for repeated measurements. For each concentration, a Tukey test was used to identify differences between groups. Responses to U46619, bradykinin, and SNP were presented as relative change in vessel diameter from baseline. For the analysis of resting arteriole diameters, HPLC-based assays, PCR, DHE and immunostainings, one-way analysis of variance (ANOVA), and Tukey’s test were used to identify differences between groups. Data are expressed as mean ± SE. The significance level was set at 0.05. All data were analyzed using GraphPad PRISM statistical software (Version 9, GraphPad Inc., San Diego, CA, USA).

## 3. Results

### 3.1. Responses of Retinal Arterioles

Initial luminal diameter of retinal arterioles incubated with physiologically concentrated glucose solution (NG, 5.5 mM D-glucose), with NG + sulodexide 50 µg/mL, with highly concentrated glucose solution (HG, 25 mM D-glucose), and with HG + sulodexide 50 µg/mL was 91 ± 8 µm, 94 ± 13 µm, 96 ± 10 µm, and 95 ± 8 µm, respectively (*p* > 0.05, one-way ANOVA and Tukey’s test). After development of myogenic tone, luminal diameter was 48 ± 5 µm, 47 ± 8 µm, 54 ± 5 µm, and 52 ± 5 µm in vessels incubated with NG, NG + sulodexide 50 µg/mL, HG, and HG + sulodexide 50 µg/mL, respectively (*p* > 0.05, one-way ANOVA and Tukey’s test). The thromboxane mimetic, U46619, elicited concentration-dependent vasoconstrictor responses, which did not differ between the groups (Figure 1A), indicating that neither HG nor sulodexide had an influence on vascular smooth muscle contractility. Likewise, the endothelium-independent vasodilator sodium nitroprusside (SNP) produced comparable responses in all groups (Figure 1B), suggesting that the ability of the vascular smooth muscle to dilate was not affected by HG or sulodexide. In contrast, responses to the endothelium-dependent vasodilator, bradykinin, were attenuated in vessels that had been incubated with highly concentrated glucose solution (HG, 25 mM D-glucose) compared to vessels incubated in physiologically concentrated glucose solution (NG, 5.5 mM D-glucose) (Figure 1C), which is indicative of endothelial dysfunction. Remarkably, when vessels incubated with HG solution were simultaneously exposed to sulodexide 50 µg/mL, vasodilator responses to bradykinin were similar to those of vessels incubated with NG solution and NG + sulodexide 50 µg/mL (Figure 1C).

To test whether sulodexide concentration-dependently prevented hyperglycemia-induced endothelial dysfunction, retinal arterioles were incubated with three different concentrations of sulodexide (50 µg/mL, 5 µg/mL, and 0.5 µg/mL). None of the concentrations affected SNP-induced vasodilation (Figure 2A). Remarkably, at higher concentrations (50 µg/mL and 5 µg/mL), sulodexide completely prevented hyperglycemia-induced endothelial dysfunction, reflected in retained bradykinin-induced responses (Figure 2B). At a sulodexide concentration of 0.5 µg/mL, a partial prevention of hyperglycemia-induced endothelial dysfunction was seen because responses at 10^−8^ M and 10^−7^ M were markedly weaker than in the NG group but stronger than in the HG group (Figure 2B).

### 3.2. Levels of Reactive Oxygen Species

DHE fluorescence intensity (Figure 3) in retinal arterioles incubated with highly concentrated glucose solution (HG) was increased by more than twofold compared to vessels incubated with normally concentrated glucose solution (NG) and with NG + 50 µg/mL sulodexide. Notably, the increase in fluorescent intensity in vessels incubated with HG was prevented when sulodexide 50 µg/mL was added to the medium (Figure 3).

### 3.3. Sulodexide as an ROS or RNS Scavenger

We did not observe considerable direct antioxidant effects of sulodexide with respect to scavenging of superoxide, peroxynitrite, and derived free radicals, as well as hydrogen peroxide (Figure 4). Although no effect at all was found regarding direct decomposition of superoxide anion radicals (Figure 4A), there was a minor trend of peroxynitrite or derived free radicals (^•^NO_2_ and ^•^OH) scavenging at the highest sulodexide concentration of 1 mM, which, however, represents a clearly supra-pharmacological concentration (Figure 4B). The highest concentration of sulodexide also caused a significant, although marginal, decrease in ebselen-oxide, indicating the moderate hydrogen peroxide scavenging activity of sulodexide (Figure 4C).

### 3.4. Messenger-RNA Expression Levels in Isolated Retinal Arterioles

Analysis of mRNA expression for the regulatory proteins SIRT1 and FOXO-1 revealed that mRNA levels for FOXO1, a transcription factor involved in the regulation of insulin signaling, were significantly elevated in the NG + Sulo and in the HG group compared to the NG group (Figure 5A). For SIRT1, a nicotinamide adenine dinucleotide (NAD+)-dependent histone deacetylase which is involved in the regulation of various physiological processes, mRNA was elevated only by tendency in the NG + Sulo and in the HG group (Figure 5A). Messenger RNA expression for the hypoxic genes, HIF-1α and VEGF-A, was similar in all groups. In the HG group, mRNA levels of both hypoxic genes were only negligibly elevated (Figure 5B). Although mRNA expression for the inflammatory genes NFκB, RANTES, and MCP-1 was similar in all groups, mRNA for IL-6 was markedly elevated in the HG group compared to the NG and the NG + Sulo group, but not to the HG + Sulo group (Figure 5C). Remarkably, mRNA expression for the pro-oxidants NOX2 and NOX4 showed no significant group-specific differences. However, NOX5 mRNA expression was significantly increased in the HG group compared to the NG and NG + Sulo group (Figure 5D). There were no group-specific differences in mRNA expression for the antioxidant genes SOD1, SOD2, SOD3, CAT, GPX1, and HO-1 (Figure 5E).

### 3.5. Expression of NOX Isoforms

Immunostainings with primary antibodies directed against NOX2, NOX4, and NOX5 were used to detect glucose- and sulodexide-dependent effects on NOX protein expression. NOX2 expression did not differ significantly between the groups. In the HG group, immunoreactivity for NOX2 was increased only negligibly (Figure 6A). In contrast, NOX4 immunoreactivity was markedly increased in the HG group compared to the NG group (*p* < 0.001) and to the group exposed to NG + sulodexide 50 µg/mL (*p* < 0.0001) (Figure 6B). Remarkably, in retinal arterioles incubated with HG + sulodexide 50 µg/mL, no increase in NOX4 immunoreactivity was observed (*p* < 0.001, HG versus HG + Sulo) (Figure 6B). Quantification of immunoreactivity against NOX5 revealed a significantly increased fluorescent signal in the HG group compared to all other groups (*p* < 0.05, HG versus NG; *p* < 0.01, HG versus NG + sulodexide 50 µg/mL and *p* < 0.01; HG versus HG + sulodexide 50 µg/mL) (Figure 6C).

## 4. Discussion

There are several major new findings in this study. First, sulodexide prevented hyperglycemia-induced endothelial dysfunction in porcine retinal arterioles in a concentration-dependent manner. Second, sulodexide reduced oxidative stress in the wall of hyperglycemia-exposed retinal arterioles. However, we could not observe considerable direct antioxidant properties of sulodexide with respect to scavenging of superoxide, peroxynitrite, and derived free radicals, as well as hydrogen peroxide. Third, sulodexide prevented hyperglycemia-induced overexpression of the pro-oxidant enzymes, NOX4 and NOX5, in the vascular wall.

Our in vitro model is based on a study by Hein et al., who observed acute hyperglycemia-induced endothelial dysfunction in porcine retinal arterioles after two hours of glucose-incubation in vitro [32]. It is noteworthy that this method induced similar endothelial dysfunction in direct comparison to a streptozotocin-induced diabetes model [32]. The domestic pig (lat. *sus scrofa domesticus*) was selected as an animal model for the present study because the porcine retina is highly comparable to the human retina in terms of anatomy, vascularization, and photoreceptor distribution [33,34].

The integrity of the vascular endothelium is essential for adequate retinal perfusion. It is well known that sustained hyperglycemia is largely responsible for endothelial damage in diabetic retinopathy [5]. Hyperglycemia-induced endothelial damage has already been observed in multiple vascular beds, such as rat aortic rings [10], human retinal endothelial cells [21], and porcine retinal arterioles [32]. Using functional vascular studies, the present study has demonstrated a significant endothelial dysfunction of retinal arterioles after only two hours of incubation with HG (25 mM D-glucose), which was reflected in a reduction of endothelium-dependent vasodilation induced by the endothelium-dependent vasodilator bradykinin. In contrast, no group-dependent differences were found in endothelium-independent vasodilation that was elicited by SNP, suggesting that vascular smooth muscle function was not affected by HG exposure. This finding is consistent with previous reports in which diabetes mellitus had no effect on SNP-induced vasodilation of retinal arterioles in a rat and pig model [35].

Strikingly, sulodexide protected retinal arterioles from HG-induced endothelial dysfunction in the present study. In accordance with our findings, sulodexide was previously shown to exert endothelium-protective effects in various organs and species, such as in small mesenteric arteries of diabetic rats, HG-incubated aortic and retinal endothelial cell cultures, human umbilical venous endothelial cells, mouse kidneys, mouse myocardium, and rat aortic rings, and, in clinical studies, on diabetic nephropathy [6,16,18,21,36,37,38,39]. Nevertheless, the underlying mechanisms are largely unknown so far.

In the present work, concentration-dependence studies were conducted using three different sulodexide concentrations (50 µg/mL, 5 µg/mL, and 0.5 µg/mL). Remarkably, all three sulodexide concentrations exerted protective effects on endothelial function, although the lowest concentration (0.5 µg/mL) only partially prevented HG-induced endothelial dysfunction. Testing the efficacy of low concentrations of sulodexide is particularly important in terms of potential clinical applicability, because plasma concentrations of sulodexide typically range between 0.5 µg/mL and 20 µg/mL [40,41,42]. Remarkably, significant endothelial protective effects in ischemic and/or hyperglycemia-exposed human umbilical venous endothelial cells have already been observed at sulodexide concentrations of 0.025 to 0.05 µg/mL [37,39]. A sulodexide concentration of 50 µg/mL showed the strongest antioxidant effect and has previously been described by other research groups [19,43]. Taken together, the optimal dose for sulodexide application in diabetic retinopathy remains to be determined. Furthermore, the optimal root of application, as shown for other substances that exert protective properties on retinal cells, needs to be reconsidered for each of the stages of diabetic retinopathy [44].

Oxidative stress is a key mechanism in the development of endothelial dysfunction in a variety of retinal pathologies [24,45,46]. Under hyperglycemic conditions, elevated ROS levels were shown to contribute to endothelial dysfunction in bovine retinal endothelial cells and pericytes [47], human umbilical venous endothelial cells [32], and human retinal endothelial cells [6]. Previous studies demonstrated that sulodexide protected glucose-exposed and senescent human umbilical venous endothelial cells from excessive ROS generation [19,37,43]. Of note, the present study is the first to report that sulodexide exposure prevents hyperglycemia-induced ROS elevation as well as endothelial dysfunction in retinal arterioles, suggesting that sulodexide targets ROS, which results in retainment of endothelial function. The mechanisms of action may involve direct antioxidant effects of sulodexide or indirect effects, such as up- or downregulation of anti- or pro-oxidant enzymes.

By using three independent methods, we found only negligible antioxidant activity of sulodexide (no scavenging of superoxide and marginal decomposition of peroxynitrite and derived free radicals, as well as hydrogen peroxide only at very high supra-pharmacological concentrations), suggesting that the substance may exert its effects by influencing other redox enzymes.

Using quantitative PCR, the expression of various regulatory, hypoxic, inflammatory, pro-oxidant, and antioxidant genes was investigated at the mRNA level. Interestingly, mRNA expression for FOXO1, a transcription factor involved in the regulation of insulin signaling; for IL-6, a pro-inflammatory cytokine; and for NOX5, a pro-oxidant enzyme, was elevated in retinal arterioles exposed to hyperglycemia, indicative of an inflammatory and pro-oxidant state in the vascular wall. Surprisingly, there have been no publications investigating a possible effect of sulodexide on the expression of NOX enzymes as major ROS producers. In diabetes, the superoxide anion (O_2_^•−^), is of particular importance, as it leads to a decreased nitric oxide bioavailability and, thus, significantly triggers endothelial dysfunction. NOX2, NOX4, and NOX5 have already been associated with the development of retinal vasculopathy in diabetic retinopathy and are also known to be expressed in human endothelial cells [13,48,49]. Because various epigenetic pathways have been implicated in the regulation of NOX expression, we tested the expression of NOX2, NOX4, and NOX5 on the protein level also. Intriguingly, immunoreactivity not only to NOX5 but also to NOX 4 was increased in retinal arterioles exposed to hyperglycemia, suggesting glucose-dependent upregulation of NOX4 and NOX5. Notably, sulodexide prevented hyperglycemia-induced overexpression of NOX4 and NOX5, suggesting that this may be a mechanism that prevents excessive ROS generation and endothelial dysfunction.

Previous studies have shown that under hyperglycemic and hypoxic conditions, increased NOX4 activity contributed to blood–retinal barrier damage and to neovascularization by inducing vascular endothelial growth factor (VEGF) expression [50]. Although numerous clinical and experimental studies have linked increased NOX4 expression to the pathogenesis of diabetic retinopathy [51] and diabetic nephropathy [52], the exact pathophysiological mechanisms remain unclear. The NOX5 isoform has rarely been studied because of its lack of expression in mouse retinal tissue. In the present study, NOX5 mRNA and protein expression were markedly increased in porcine retinal arterioles under hyperglycemic conditions. The role of NOX5 in pathologically increased cell permeability and neovascularization in diabetic retinopathy is the subject of current research [48]. In addition, increased expression of human NOX5 has been observed in endothelial cells and smooth muscle cells of transgenic mice in association with renal inflammatory processes [53]. Upregulation of NOX5 in diabetic nephropathy in human podocytes has been associated with filter barrier disruption [54]. Surprisingly, no high glucose- or sulodexide- associated effects on NOX2 expression were observed in the present study. Remarkably, sulodexide prevented hyperglycemia-induced NOX4 and NOX5 expression in the vascular wall, suggesting that this may be one of the mechanisms preventing ROS generation. One question which remains to be solved is the lack of NOX4 mRNA level changes by hyperglycemia in our model, despite changes on the protein level.

In the context of in vivo studies, authors usually observed a concordant change in NOX expression at the mRNA and protein levels [24,45,46]. An expression change at the transcriptional level seems to occur mainly in a prolonged diabetic milieu, as already shown in diabetic rat models, in which increased mRNA expression of the NOX4 and p22phox subunits was seen in the kidney after several weeks of exposure [50,55]. Interestingly, in human mesangial cell cultures, increased ROS levels have been observed after three days of exposure to highly concentrated glucose solution, but an increase in mRNA expression for pro-oxidant enzymes occurred several days later. Surprisingly, these early changes were completely reversible by simultaneous incubation with a NOX inhibitor, so the authors concluded that enzyme expression may be upregulated secondarily to initial enzyme activation [55]. It is reasonable to assume that two hours of incubation is too short to induce profound changes on the transcription level, which may explain why only minimal changes in mRNA expression levels were seen in response to hyperglycemia in our study. One possibility is that sulodexide decreased translation of NOX mRNA in this context. However, post-transcriptional effects may also be caused by noncoding RNA, such as micro-RNAs, which are able to regulate gene expression through direct mRNA binding and, thus, may also represent an interesting therapeutic target. The exact influences of micro-RNAs on NOX gene expression are unclear so far, but it has already been shown that miRNA-25 binds directly to the human NOX4 gene and that downregulation of this led to increased ROS production under diabetic conditions [56,57]. It is worth mentioning that NOX inhibition can be achieved in different signaling pathways. In addition to nonspecific mechanisms, an effect on protein kinase C, which can inhibit NOX activation by blocking protein phosphorylation [58], is of importance. Moreover, a glucose- and sulodexide-associated effect on individual NOX subunits—specifically the important catalytic subunit gp91-phox, but also interactions between different subunits—is conceivable [58]. In this context, it would be possible that mRNA expression is altered only for individual subunits but not with respect to the overall gene. Furthermore, an influence of sulodexide on proteasomal degradation of NOX enzymes could be presumed. In vivo, diabetes can cause a dysregulated activity of the ubiquitin–proteasome complex, which is responsible for the degradation of many intracellular enzymes. This induces an enhanced vascular inflammatory response in which the NFkB pathway plays a major role [59]. Of great importance for the degradation of damaged and cell-damaging proteins is the process of autophagy. Increased oxidative stress removes the control of autophagy metabolism; consequently, for example, there is an accumulation of pro-oxidant enzymes, such as NOX. It could be possible that sulodexide directly or indirectly affects autophagy under oxidative stress conditions.

## 5. Conclusions

To conclude, hyperglycemia caused marked endothelial dysfunction and elevated ROS levels in porcine retinal arterioles after only two hours. Strikingly, sulodexide concentration-dependently provided protection from hyperglycemia-induced endothelial dysfunction via the blockade of ROS formation. Although sulodexide itself had only negligible antioxidant properties, it prevented hyperglycemia-induced overexpression of the pro-oxidant redox enzymes NOX4 and NOX5. From a clinical point of view, sulodexide may represent a therapeutic approach to reduce oxidative stress in the treatment of diabetic retinopathy.

## Figures and Tables

**Figure 1 antioxidants-12-00388-f001:**
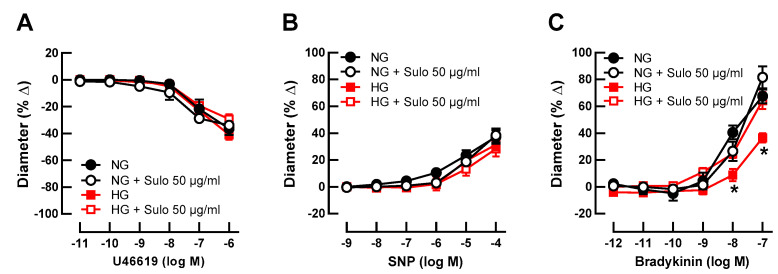
Concentration-dependent responses of porcine retinal arterioles after two hours of incubation with HG (highly concentrated glucose solution; 25 mM D-glucose), HG + Sulo 50 µg/mL (highly concentrated glucose solution + 50 µg/mL sulodexide), NG (normally concentrated glucose solution; 5.5 mM D-glucose), NG + Sulo 50 µg/mL (normally concentrated glucose solution + 50 µg/mL sulodexide) to the vasoconstrictor U46619 (**A**), the endothelium-independent vasodilator SNP (**B**), and to the endothelium-dependent vasodilator bradykinin (**C**). Data are presented as mean ± SE (*n* = 8 per concentration and group; * *p* < 0.05, HG versus all other groups).

**Figure 2 antioxidants-12-00388-f002:**
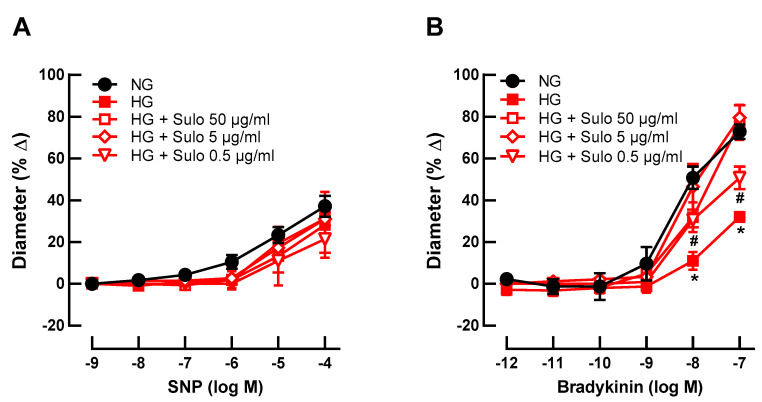
Concentration-dependence studies for sulodexide. Vascular reactivity of retinal arterioles to the endothelium-independent vasodilator SNP (**A**) and to the endothelium-dependent vasodilator bradykinin (**B**) after incubation with highly concentrated glucose solution (HG) and different sulodexide concentrations for two hours: 50 µg/mL, 5 µg/mL, and 0.5 µg/mL. Vessels incubated with normally concentrated glucose solution (NG) served as controls. Data are expressed as mean ± SE (*n* = 8 per concentration and group; * *p* < 0.05, HG versus all other groups; # *p* < 0.05, HG + Sulo 0.5 µg/mL versus NG).

**Figure 3 antioxidants-12-00388-f003:**
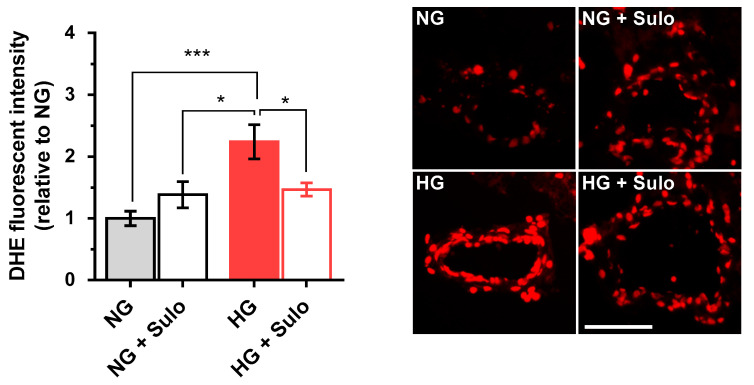
DHE staining of retinal arterioles in cryosections after two hours of incubation with HG (highly concentrated glucose solution; 25 mM D-glucose), HG + Sulo 50 µg/mL (highly concentrated glucose solution + 50 µg/mL sulodexide), NG (normally concentrated glucose solution; 5.5 mM D-glucose), and NG + Sulo 50 µg/mL (NG + 50 µg/mL sulodexide). The microphotographs show representative stainings of retinal arteriole cross-sections for the individual groups. Data are expressed as mean ± SE (*n* = 8 per group; * *p* < 0.05, *** *p* < 0.001). Scale bar = 50 µm.

**Figure 4 antioxidants-12-00388-f004:**
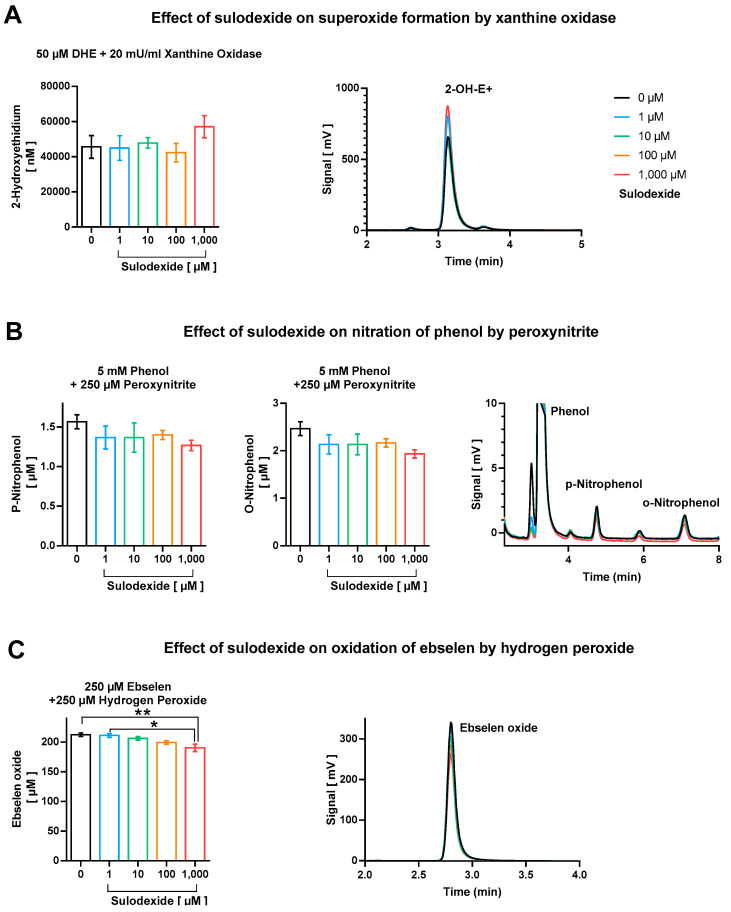
Direct antioxidant properties of sulodexide tested on (**A**) superoxide formation by xanthine oxidase (20 mU/mL)/1 mM hypoxanthine with hydroethidine (50 µM), (**B**) nitration of phenol (5 mM) by peroxynitrite (250 µM), and (**C**) oxidation of ebselen (250 µM) by hydrogen peroxide (250 µM). Representative chromatograms are shown besides the quantification bar graphs. Data are expressed as mean ± SE (*n* = 3–5 per group; * *p* < 0.05; ** *p* < 0.01).

**Figure 5 antioxidants-12-00388-f005:**
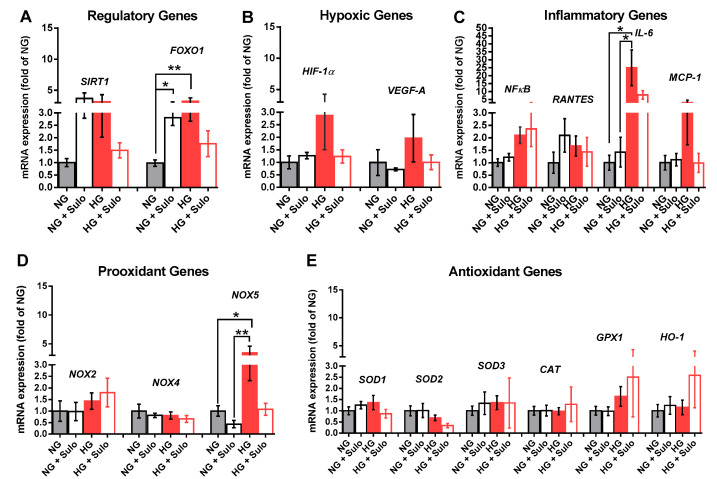
Messenger RNA expression of regulatory genes (*SIRT1*, *FOXO1*, (**A**)), hypoxic genes (*HIF-1α*, *VEGF-A*, (**B**)), inflammatory genes (*NFκB*, *RANTES*, *IL-6*, *MCP-1*, (**C**)), pro-oxidant redox genes (*NOX2*, *NOX4*, *NOX5*, (**D**)), and antioxidant redox genes (*SOD1*, *SOD2*, *SOD3*, *CAT*, *GPX1*, *HO-1*, (**E**)) after incubation for two hours with HG (highly concentrated glucose solution; 25 mM D-glucose), HG + Sulo 50 µg/mL (highly concentrated glucose solution + 50 µg/mL sulodexide), NG (normally concentrated glucose solution; 5.5 mM D-glucose), and NG + Sulo 50 µg/mL (NG + 50 µg/mL sulodexide). Data are expressed as mean ± SE (*n* = 8 per group; * *p* < 0.05, ** *p* < 0.01). In each case, mRNA expression levels were normalized to the NG group.

**Figure 6 antioxidants-12-00388-f006:**
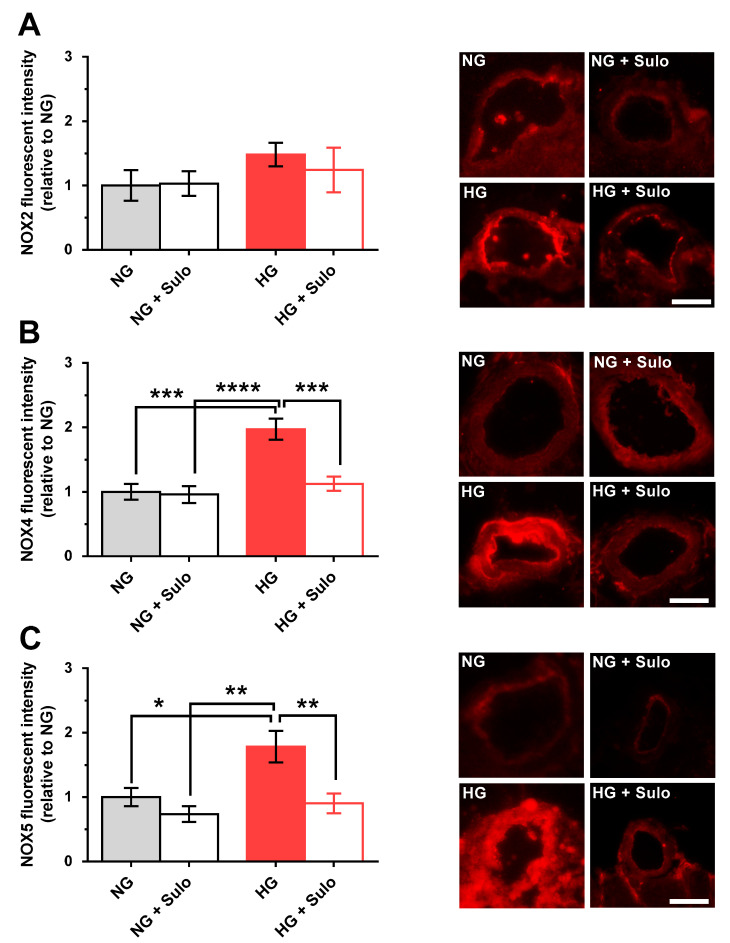
Immunofluorescence micrographs of retinal arteriole cross-sections (cryosections) stained with a primary antibody (Abcam, Waltham, MA, USA, dilution 1:200, incubation time: 2 h at RT) targeted against NOX2 (ab80508), NOX4 (ab154244), and NOX5 (ab191010) and a rhodamine Red-X-coupled secondary antibody (Dianova GmbH, Hamburg, Germany, 111-295-003, dilution 1:200, incubation time: 2 h at RT) (red color). Quantification of fluorescent intensities in the vascular wall of retinal arterioles normalized to NG. (**A**) Immunoreactivity to NOX2 was similar in all groups. (**B**) In contrast, immunoreactivity to NOX4 was elevated in the HG group. (**C**) Likewise, immunoreactivity to NOX5 was increased in the HG group. Data are expressed as mean ± SE (*n* = 8 per group; * *p* < 0.05, ** *p* < 0.01, *** *p* < 0.001, **** *p* < 0.0001). Scale bar = 50 µm.

**Table 1 antioxidants-12-00388-t001:** Primer sequences used for qPCR.

Gene	Forward	Reverse	Accession Number
*SIRT1*	GAGAAGGAAACAATGGGCCG	ACCAAACAGAAGGTTATCTCGGT	NM_001145750.2
*FOXO1*	CGGCAGGCTGGAAGAATTCAA	CTCCCTCTGGGTTGAGCATC	NM_214014.3
*HIF-1α*	CTCCATTGCCTGCCTCTGAA	TGGGACTGTTAGGCTCAGGT	NM_001123124.1
*VEGF-A*	CGAGGCAAGAAAATCCCTGT	GCGAGTCTGTGTTTTTGCAGG	NM_214084.1
*NFκB*	AACAACCCCTTCCAAGTTCCC	GCACGGTTGTCAAAGATGGG	NM_001114281.1
*RANTES*	ATGGCAGCAGTCGTCTTTATC	TGCACGAGTTCAGGCTCAAG	NM_001129946.1
*IL-6*	AGACCCTGAGGCAAAAGGGAAA	TCAGGTGCCCCAGCTACAT	NM_214399.1
*MCP-1*	CTTGAATCCTCATCCTCCAGCA	CTGGAGAATTAATTGCATCTGGC	NM_214214.1
*NOX2*	CACTTCACGCCACGATTCAC	TTGACTCGGGCGTTCACAC	NM_214043.2
*NOX4*	GTCCCAGTGTGTCTGCGTTAG	TCTCGAAATCGTTCTGTCCAGTC	XM_013979249.2
*NOX5*	AAGAGTCCTTCTTTGCTGAGAGA	CAGCCAGTGAACAGCACTGA	XM_021100544.1
*SOD1*	GGGCACCATCTACTTCGAGC	CTGCACTGGTACAGCCTTGT	NM_001190422.1
*SOD2*	GGCCTACGTGAACAACCTGA	AATTCCCCTTTGGGTTCCCC	NM_214127.2
*SOD3*	GAAGAGCTGGAAAGGTGCCC	ATCTCCGTCACTTTGGCCTG	XM_021100523.1
*CAT*	GCTTCAACAGTGCCAACGAA	ACTGAAGTTCTTGACCGCTTTC	XM_021081498.1
*GPX1*	AGTTTGGACATCAGGAAAATGCC	AGCATGAAGTTGGGCTCGAA	NM_214201.1
*HO-1*	TGATGGCGTCCTTGTACCAC	GACCGGGTTCTCCTTGTTGT	NM_001004027.1
*ACTB*	TGGACTACCTCCTGTCTGCT	CCTAGGGGTGGGTTTCTGTG	XM_021086047.1

## Data Availability

The data presented in this study are available on request from the corresponding author.

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
