# Peer review of "Sulodexide Prevents Hyperglycemia-Induced Endothelial Dysfunction and Oxidative Stress in Porcine Retinal Arterioles"

_antioxidants, 2023, doi:10.3390/antiox12020388_

Round 1

Reviewer 1 Report

The current manuscript aims to report that sulodexide prevents from hyperglycemia-induced endothelial dysfunction and oxidative stress in porcine retinal arterioles. Although the topic is significant in the field of development of antioxidants for the treatment of diabetic retinopathy, there are several issues that definitely require the authors’ attention to improve the quality of this particular manuscript before further consideration for publication in a high-quality journal “Antioxidants”.

Specific comments:

1.         In fact, investigators have reported the use of sulodexide to inhibit retinal neovascularization in an animal model of oxygen-induced retinopathy (DOI: 10.5483/bmbrep.2014.47.11.009). The authors should clarify the academic novelty by distinguishing their work from this earlier report.

2.         In this work, the disease model is a key to test the antioxidants. The audiences are curious about whether the animal model of porcine retinal arterioles can be successfully induced by incubation with highly concentrated glucose solution? The authors should provide solid evidences to support the reasonableness of biological studies.

3.         The audiences are unaware of the underlying reason of using U46619, SNP, and bradykinin in Section 3.1 Responses of retinal arterioles. Please specify.

4.         Why the authors selected the concentration of 50 μg/ml of sulodexide for testing? If the sulodexide has dose-dependent effect, the authors should consider the increase of the sulodexide concentration. Please clarify the necessity of this critical experimental condition.

5.         As stated by the authors, at higher concentrations (50 μg/ml and 5 μg/ml) sulodexide completely prevented hyperglycemia-induced endothelial dysfunction. But, as shown in Figure 2B, the low sulodexide concentration (i.e., 5 μg/ml) exhibited better performance to prevent hyperglycemia-induced endothelial dysfunction than its counterpart of 50 μg/ml sulodexide. The data presentation seems contradictory to the authors’ claim. Please clarify this important issue.

6.         As shown in Figure 3, the DHE fluorescent intensity in the NG+Sulo group is higher than that in the NG group. Based on this observation, the audiences may misleadingly consider the pro-oxidant effect of Sulo. Furthermore, please carefully check the luminal vessel diameter in the DHE fluorescent images shown in Figure 3 since the results of diameter seem not well matched with the data presented in Figure 1 and Figure 2.

7.         Figure 6 shows immunofluorescence micrographs of retinal arteriole cross-sections (cryosections). However, it is unclear to see whether the vessel diameter will be affected by either the hyperglycemia-induced endothelial dysfunction or sulodexide treatment? Please specify. Furthermore, please add scale bars to fluorescent imaging data shown in Figure 6.

8.         In this work, the prevention of oxidative stress in porcine retinal arterioles by sulodexide was reported. But, the audiences are curious about whether the authors also performed the tests to examine the endothelial dysfunction following sulodexide treatment? Please improve this important issue.

9.         As stated by the authors, oxidative stress is a key mechanism in the development of endothelial dysfunction in a variety of retinal pathologies. Although some literature reports are cited in this manuscript, one important example (DOI: 10.1021/acsnano.2c05824) should also be illustrated to balance the scientific viewpoint and attract more attention from audiences. If possible, the authors are highly recommended to consider the inclusion of this relevant paper in the reference list to update and deepen the Discussion section of the article content.

Author Response

Dear Reviewer 1, thank you very much for your comments and suggestions. Please find below our responses to these comments.

Kindest regards

Alice Dauth

Point 1

In fact, investigators have reported the use of sulodexide to inhibit retinal neovascularization in an animal model of oxygen-induced retinopathy (DOI: 10.5483/bmbrep.2014.47.11.009). The authors should clarify the academic novelty by distinguishing their work from this earlier report.

 Response to Point 1:

While Jo et al. (2014) reported on an anti-angiogenetic effect of sulodexide in a mouse model of oxygen-induced retinopathy [1], we used an in vitro model of acute hyperglycemia, which may be suitable to detect acute molecular and functional changes but not chronic changes, such as new vessel formation. It needs to be pointed out that oxygen-induced retinal retinopathy is a disease occurring in preterm born children, who receive supplemental oxygen, whereas diabetic retinopathy occurs in diabetic patients, which are typically older. Hence, the pathophysiology differs considerably between these two diseases. We now cite the publication by Jo et al in the introduction (line 86).

Point 2

In this work, the disease model is a key to test the antioxidants. The audiences are curious about whether the animal model of porcine retinal arterioles can be successfully induced by incubation with highly concentrated glucose solution? The authors should provide solid evidences to support the reasonableness of biological studies.

Response to Point 1:

The high glucose concentration we used (25 mmol/l = 450 mg/dl) can be reached in some diabetic patients. Our in-vitro-model was based on a study by Hein et al., who investigated acute hyperglycemia-induced damage in porcine retinal vessels after two hours of glucose incubation in-vitro. It is noteworthy that this method induced similar endothelial dysfunction in direct comparison to a streptozotocin-induced diabetes model [2]. The domestic pig (lat. sus scrofa domesticus) was selected as an animal model for the present research, because the porcine retina is highly comparable to the human retina in terms of anatomy, vascularization and photoreceptor distribution [3, 4]. Hein et al. published various studies that disclosed similarities in the vasoreactivity and its underlying signaling mechanisms between human and porcine retinal arterioles. [5, 6].

Point 3

The audiences are unaware of the underlying reason of using U46619, SNP, and bradykinin in Section 3.1 Responses of retinal arterioles. Please specify.

Response to Point 3

According to the suggestion of the Reviewer, we now included explanations regarding the reason of these substances in Section 3.1.

Point 4

Why the authors selected the concentration of 50 μg/ml of sulodexide for testing? If the sulodexide has dose-dependent effect, the authors should consider the increase of the sulodexide concentration. Please clarify the necessity of this critical experimental condition.

Response to Point 4

A sulodexide concentration of 50 µg/ml is already supratherapeutic. In patients on medication with sulodexide, blood concentrations of 0.5 µg/ml and 20 µg/ml have been reported, while endothelium-protective and antioxidant effects could already be demonstrated at concentrations lower than 50 µg/ml by other research groups [7-9]. We also tested the efficacy of lower concentrations than 50 µg/ml, because we find this particularly important in terms of potential clinical applicability [10-13]. The fact that hyperglycemia-induced vascular endothelial dysfunction was completely prevented at a lower concentration of sulodexide (5 µm/ml) suggests that 50 µg/ml was already supramaximal and no further protective effect can be expected when using even higher concentrations.

Point 5

As stated by the authors, at higher concentrations (50 μg/ml and 5 μg/ml) sulodexide completely prevented hyperglycemia-induced endothelial dysfunction. But, as shown in Figure 2B, the low sulodexide concentration (i.e., 5 μg/ml) exhibited better performance to prevent hyperglycemia-induced endothelial dysfunction than its counterpart of 50 μg/ml sulodexide. The data presentation seems contradictory to the authors’ claim. Please clarify this important issue.

 Response to Point 5

In fact, the concentration-response curves with 50 µg/ml and 5 µg/ml of sulodexide were quite similar and did not differ statistically from each other. At a concentration of 10-8 M bradykinin, the response is by tendency lower in the 50 µg/ml than in the 5 µg/ml sulodexide group. Since this difference is not statistically significant, we did not discuss it. One may speculate that at supratherapeutic concentrations there may be even toxic effects of a given substance. However, we would like not to speculate too much when the effect is not significant.

Point 6

As shown in Figure 3, the DHE fluorescent intensity in the NG+Sulo group is higher than that in the NG group. Based on this observation, the audiences may misleadingly consider the pro-oxidant effect of Sulo. Furthermore, please carefully check the luminal vessel diameter in the DHE fluorescent images shown in Figure 3 since the results of diameter seem not well matched with the data presented in Figure 1 and Figure 2.

Response to Point 6

In Figure 3, DHE fluorescence is by tendency higher in the NG+Sulo than in the NG group. However, these two groups do not differ in a statistical sense from each other. We would not like to interpret data in the manuscript, which do not differ significantly. We understand that one may expect that the fluorescent intensity should be lower in the NG+Sulo than in the NG group. However, reactive oxygen species, which at low concentrations are important for cellular functions, are generated by different sources in the cell, e.g., by NOX enzymes, by xanthine oxidase and by mitochondrial enzymes. It is possible that under normoglycemic conditions, sulodexide does not prevent formation of reactive oxygen species, which would be even desirable. It is possible that sulodexide prevents formation of some NOX enzymes under hyperglycemic conditions only.

The diameter of retinal arterioles differs depending on various factors, e.g., vascular tone at isolation and the distance from the optic nerve. The arteriole diameter reduces from the center to the periphery. Therefore, it is difficult to compare the diameter in histological sections. Diameter is better presented in videomicroscopic measurements where luminal arteriole pressure is standardized. We now present the diameter before and after development of myogenic tone measured with videomicroscopy (Results 3.1.). In Figure 1 and 2, vascular reactivity is presented as change in percent, therefore, not taking into account the real diameter.

Point 7

Figure 6 shows immunofluorescence micrographs of retinal arteriole cross-sections (cryosections). However, it is unclear to see whether the vessel diameter will be affected by either the hyperglycemia-induced endothelial dysfunction or sulodexide treatment? Please specify. Furthermore, please add scale bars to fluorescent imaging data shown in Figure 6.

Response to Point 7

We answered the first part of the question regarding the diameter in our previous answer to point 6.

According to the Reviewer’s suggestion, we added scale bars to the pictures of Figure 6.

Point 8

In this work, the prevention of oxidative stress in porcine retinal arterioles by sulodexide was reported. But the audiences are curious about whether the authors also performed the tests to examine the endothelial dysfunction following sulodexide treatment? Please improve this important issue.

Response to Point 8

The effects of sulodexide on endothelial dysfunction were investigated by functional vascular studies.

As shown in Figure 1, one group received sulodexide alone under normoglycemic conditions to test wheter sulodexide induced changes in vascular function.

Point 9

As stated by the authors, oxidative stress is a key mechanism in the development of endothelial dysfunction in a variety of retinal pathologies. Although some literature reports are cited in this manuscript, one important example (DOI: 10.1021/acsnano.2c05824) should also be illustrated to balance the scientific viewpoint and attract more attention from audiences. If possible, the authors are highly recommended to consider the inclusion of this relevant paper in the reference list to update and deepen the Discussion section of the article content.

Response to Point 9

According to the suggestion of the Reviewer, we added a statement in the Discussion (lines 410-412) citing the recommended article.

  1. Jo, H.; Jung, S. H.; Kang, J.; Yim, H. B.; Kang, K. D., Sulodexide inhibits retinal neovascularization in a mouse model of oxygen-induced retinopathy. BMB Rep 2014, 47, (11), 637-42.
  2. Hein, T. W.; Xu, W.; Xu, X.; Kuo, L., Acute and Chronic Hyperglycemia Elicit JIP1/JNK-Mediated Endothelial Vasodilator Dysfunction of Retinal Arterioles. Investigative ophthalmology & visual science 2016, 57, (10), 4333-40.
  3. Guduric-Fuchs, J.; Ringland, L. J.; Gu, P.; Dellett, M.; Archer, D. B.; Cogliati, T., Immunohistochemical study of pig retinal development. Molecular vision 2009, 15, 1915-28.
  4. Sanchez, I.; Martin, R.; Ussa, F.; Fernandez-Bueno, I., The parameters of the porcine eyeball. Graefes Arch Clin Exp Ophthalmol 2011, 249, (4), 475-82.
  5. Hein, T. W.; Potts, L. B.; Xu, W.; Yuen, J. Z.; Kuo, L., Temporal development of retinal arteriolar endothelial dysfunction in porcine type 1 diabetes. Investigative ophthalmology & visual science 2012,53, (13), 7943-9.
  6. Hein, T. W.; Rosa, R. H., Jr.; Yuan, Z.; Roberts, E.; Kuo, L., Divergent roles of nitric oxide and rho kinase in vasomotor regulation of human retinal arterioles. Investigative ophthalmology & visual science 2010, 51, (3), 1583-1590.
  7. Ciszewicz, M.; Polubinska, A.; Antoniewicz, A.; Suminska-Jasinska, K.; Breborowicz, A., Sulodexide suppresses inflammation in human endothelial cells and prevents glucose cytotoxicity. Translational research : the journal of laboratory and clinical medicine 2009, 153, (3), 118-123.
  8. Suminska-Jasinska, K.; Polubinska, A.; Ciszewicz, M.; Mikstacki, A.; Antoniewicz, A.; Breborowicz, A., Sulodexide reduces senescence-related changes in human endothelial cells. Medical science monitor : international medical journal of experimental and clinical research 2011, 17, (4), Cr222-6.
  9. Gabryel, B.; Bontor, K.; Jarząbek, K.; Plato, M.; Pudełko, A.; Machnik, G.; Urbanek, T., Sulodexide up-regulates glutathione S-transferase P1 by enhancing Nrf2 expression and translocation in human umbilical vein endothelial cells injured by oxygen glucose deprivation. Arch Med Sci 2020, 16, (4), 957-963.
  10. Coccheri, S.; Mannello, F., Development and use of sulodexide in vascular diseases: implications for treatment. Drug Des Devel Ther 2013, 8, 49-65.
  11. Borawski, J.; Dubowski, M.; Pawlak, K.; Mysliwiec, M., Sulodexide induces hepatocyte growth factor release in humans. Eur J Pharmacol 2007, 558, (1-3), 167-71.
  12. Condorelli, M.; Chiariello, M.; Dagianti, A.; Penco, M.; Dalla Volta, S.; Pengo, V.; Schivazappa, L.; Mattioli, G.; Mattioli, A. V.; Brusoni, B.; et al., IPO-V2: a prospective, multicenter, randomized, comparative clinical investigation of the effects of sulodexide in preventing cardiovascular accidents in the first year after acute myocardial infarction. J Am Coll Cardiol 1994, 23, (1), 27-34.
  13. Gambaro, G.; Kinalska, I.; Oksa, A.; Pont'uch, P.; Hertlová, M.; Olsovsky, J.; Manitius, J.; Fedele, D.; Czekalski, S.; Perusicová, J.; Skrha, J.; Taton, J.; Grzeszczak, W.; Crepaldi, G., Oral sulodexide reduces albuminuria in microalbuminuric and macroalbuminuric type 1 and type 2 diabetic patients: the Di.N.A.S. randomized trial. J Am Soc Nephrol 2002, 13, (6), 1615-25.

Reviewer 2 Report

This study demonstrated that sulodexide, a drug used for venous insufficiency, prevention of venous thromboembolism and for treatment of diabetic nephropathy, can prevent in a concentration-dependent manner the hyperglycemia-induced endothelial dysfunction, as well as ROS elevation, in an in vitro model of retinal arterioles from pig. The finding of a negligible antioxidant activity of sulodexide suggests that the drug may exert its favourable vascular protective effects via redox enzymes. In particular, sulodexide prevented hyperglycemia-induced overexpression of pro-oxidant enzymes NOX4 and NOX5, indicating this could be the mechanism counteracting excessive ROS generation. The results suggest that sulodexide may become a therapeutic approach to reduce oxidative stress in the treatment of diabetic retinopathy.

The experimental design is clearly outlined, methodology accurately described, results presented in a clear sequence, and discussion well-constructed to consider all the experimental data in the context of knowledge and possible treatment of diabetic microvascular complications, in particular retinopathy.

Just a few minor formal details to be corrected in the manuscript:

 Line 97 and line 363: the scientific binomial nomenclature for domestic pig should be reported following taxonomic rules, that is as italic font, and with Genus name capitalized (Sus scrofa domesticus)

 Line 265: Figure 2 presents not only bradykinin concentration-response curve, but also that of SNP; so also the latter drug should be mentioned in legend.

 Line 349: please use point as decimal separator. Same comment also for line 388.

 Line 5: incorrectly spelling of name “ian” for “Adrian”

Author Response

Dear Reviewer 2, thank you very much for your comments and suggestions. Changes were directly made in our manuscript. 

Kindest regards

Alice Dauth

Round 2

Reviewer 1 Report

The revised version has adequately addressed most of the critiques raised by this reviewer and is now suitable for publication in "Antioxidants".